# ON THE IMPORTANCE OF DIFFICULTY CALIBRATION IN MEMBERSHIP INFERENCE ATTACKS

**Lauren Watson**[*]
University of Edinburgh

**Chuan Guo**

**Graham Cormode**
Meta AI

**Alexandre Sablayrolles**

## ABSTRACT

The vulnerability of machine learning models to membership inference attacks has received much attention in recent years. Existing attacks mostly remain impractical due to having high false positive rates, where non-member samples are often erroneously predicted as members. This type of error makes the predicted membership signal unreliable, especially since most samples are non-members in real world applications. In this work, we argue that membership inference attacks can benefit drastically from *difficulty calibration*, where an attack's predicted membership score is adjusted to the difficulty of correctly classifying the target sample. We show that difficulty calibration can significantly reduce the false positive rate of a variety of existing attacks without a loss in accuracy.

## 1    INTRODUCTION

Modern applications of machine learning often involve training models on sensitive data such as health records and personal information. Unfortunately, recent studies have found that these models can memorize their training data to a large extent, compromising the privacy of participants in the training dataset (Fredrikson et al., 2014; 2015; Shokri et al., 2017; Carlini et al., 2019). One prominent category of privacy attacks against machine learning is the so-called *membership inference attack* (Shokri et al., 2017; Yeom et al., 2018), where the adversary aims to infer the participation of an individual in the target model's training set. Such an attack is undoubtedly damaging when the status of participation itself is considered sensitive, *e.g.*, if the training dataset consists of health records of cancer patients. Moreover, the ability to infer membership can be viewed as a lower bound for the model's degree of memorization (Yeom et al., 2018), which is useful in itself as an empirical quantifier of privacy loss (Jagielski et al., 2020; Nasr et al., 2021).

The efficacy of membership inference attacks has been improved significantly since the first attempts (Salem et al., 2018; Sablayrolles et al., 2019; Leino & Fredrikson, 2020). However, the most common evaluation metric, attack accuracy, overlooks the crucial factor of the *false positive rate* (FPR) of non-members (Rezaei & Liu, 2021). Indeed, most attacks operate by first defining a membership score $s(h, \boldsymbol{z})$ for the model $h$ and a target input-label pair $\boldsymbol{z} = (\boldsymbol{x}, y)$ that measures how much $h$ memorized the sample $\boldsymbol{z}$. The attack subsequently selects a threshold $\tau$ and predicts that $\boldsymbol{z}$ is a member if and only if $s(h, \boldsymbol{z}) > \tau$. For typical choices of the membership scoring function, there is usually a large overlap in the distribution of $s(h, \boldsymbol{z})$ between members and non-members (see Figure 1a). As a result, an attack that determines the membership of $\boldsymbol{z}$ by thresholding on $s(h, \boldsymbol{z})$ will have a high FPR. This drawback renders most existing attacks unreliable since the vast majority of samples likely belong to the non-member class.

In this study, we identify the lack of *difficulty calibration* as a core contributor to the high FPR of existing attacks. Specifically, a non-member sample may have a high membership score simply because it is over-represented in the data distribution. Consequently, an attack that determines a sample is likely to be a member due to having a high score will inevitably fail on these over-represented samples. To remedy this problem, we make the acute observation that if the membership score is measured *in comparison* to a typical model trained on data drawn from the same data distribution, this difference in behavior can serve as a much more reliable membership signal. Indeed,

---

[*]Work done during an internship at Facebook. Email:lauren.watson@ed.ac.uk, {chuanguo, gcormode, asablayrolles}@fb.com

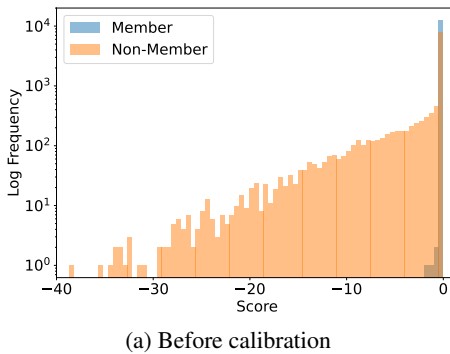
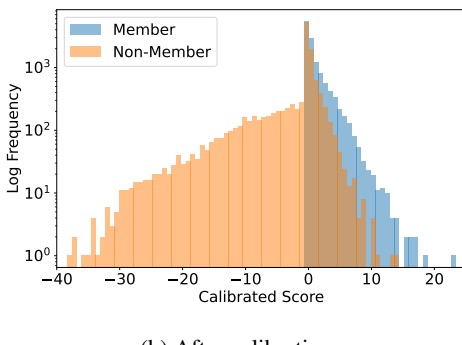

(a) Before calibration

(b) After calibration

Figure 1: Histogram of the negative loss score (*cf.* Equation 1) before and after difficulty calibration. Without calibration, the member and non-member scores overlap significantly, and it is impossible to determine a threshold that results in low FPR. After calibration, the highest scored samples mostly belong to the member class, enabling high precision and low FPR attacks.

Figure 1b shows the histogram of scores $s(h, \boldsymbol{z})$ after difficulty calibration, where the member and non-member scores have significantly better separation and low FPR is now attainable.

We propose difficulty calibration as a general technique for improving score-based membership inference attacks, and modify several membership scoring functions such as confidence (Salem et al., 2018), loss (Yeom et al., 2018), and gradient norm (Nasr et al., 2019) to construct their calibrated variants. Evaluated on a comprehensive suite of benchmark datasets, we show that calibrated attacks achieve a significantly lower FPR compared to prior work. In particular, we measure the trade-off between true positives and false positives using the *area under ROC curve* (AUC) metric, and show that difficulty calibration drastically improves this trade-off compared to uncalibrated attacks, by up to 0.10 AUC on common ML benchmarks. In addition, calibrated attacks also drastically improve the precision-recall trade-off, while remaining on-par with or better than uncalibrated attacks in terms of attack accuracy. Our results suggest that it may be important for future work to apply difficulty calibration to design more reliable and practical membership inference attacks.

## 2 BACKGROUND

*Membership inference attacks* are concerned with determining whether a given sample was part of a target model's training set. Homer et al. (2008) showed in their pioneering study that it is possible to infer an individual's presence in a complex genomic DNA mixture, which led to increased caution around releases of DNA data (Zerhouni & Nabel, 2008). Recent interest in member inference attacks was sparked by the work of Shokri et al. (2017), who introduced the *shadow models* method: an adversary trains substitute models (called *shadow models*) to mimic the behavior of the model under attack (called the *target model*). The adversary then observes the behavior of the shadow models when exposed to member and non-member samples, and uses this observation to train an attack meta-model for predicting membership on any given sample. Shokri et al. (2017) evaluated this attack on ML models trained on cloud APIs, and showed that the shadow models approach attains high levels of accuracy.

**Score-based attacks.** Yeom et al. (2018) discovered a connection between membership inference attacks and overfitting, arguing that in principle, attack accuracy can be determined by how much the target model memorizes (or overfits to) the given sample $(\boldsymbol{x}, y)$. This discovery led to a series of work on quantifying the degree of memorization via *membership scores*, which can be used for predicting that the given sample is a member when the score is high. The membership score can be computed using the loss (Yeom et al., 2018), the gradient norm (GN) (Nasr et al., 2019) or the confidence of the model's prediction (Salem et al., 2018), often yielding state-of-the-art results (Salem

et al., 2018; Choquette-Choo et al., 2021). We define these scores for the cross-entropy loss $\ell$:

$$s_{\text{loss}}(h, (\boldsymbol{x}, y)) = -\ell(h(\boldsymbol{x}), y) := \log(h(\boldsymbol{x})_y), \tag{1}$$

$$s_{\text{GN}}(h, (\boldsymbol{x}, y)) = -\|\nabla \ell(h(\boldsymbol{x}), y)\|_2, \tag{2}$$

$$s_{\text{confidence}}(h, (\boldsymbol{x}, y)) = -\max_{y'} \ \ell(h(\boldsymbol{x}), y') = \max_i \ \log(h(\boldsymbol{x})_i). \tag{3}$$

**Label-only attacks.** The above score-based attacks rely on continuous-valued predictions from the model in order to define the membership score. To counter these attacks, prior work considered obfuscating the model's output by returning only the top label or modifying the predicted values (Shokri et al., 2017; Jia et al., 2019). However, subsequent studies showed that even with only hard-label output, it is possible to define scores attaining close to state-of-the-art accuracy (Li & Zhang, 2020; Choquette-Choo et al., 2021).

**High-precision attacks.** Various forms of difficulty calibration have been considered in the context of high-precision attacks. Long et al. (2018) selected samples that differ the most in loss between the target and a set of reference models, and showed that the resulting attack has high precision even for well-generalized target models. Carlini et al. (2020) showed that privacy attacks are also possible on large-scale language models such as GPT-2 (Radford et al., 2019). Their attack operates by first generating a large number of sentences from the language model and then ranking these sentences by the (log) perplexity, with lower perplexity indicating a more plausible memorized training sample. These perplexity values are then divided by either the z-lib entropy or the perplexity given by a smaller language model to account for the sentence's rarity. In effect, only rare sentences with low perplexity can minimize the resulting score. Both attacks leverage a form of difficulty calibration by comparing the target model's loss with that of reference models, predicting membership only when the difference (or ratio) is large.

**Differential privacy as a mitigation.** Differential privacy (Dwork et al., 2006) (DP) is a powerful mathematical framework for privacy-preserving data analysis. A randomized algorithm $\mathcal{M}$ satisfies $(\epsilon, \delta)$-differential privacy if, given any two datasets $\mathcal{D}$ and $\mathcal{D}'$ that differ in at most one sample, and for any subset $R$ of the output space, we have:

$$\mathbb{P}(\mathcal{M}(\mathcal{D}) \in R) \leq \exp(\epsilon)\mathbb{P}(\mathcal{M}(\mathcal{D}') \in R) + \delta. \tag{4}$$

Under mild assumptions, DP provably protects against a variety of privacy attacks, in particular membership inference (Yeom et al., 2018; Sablayrolles et al., 2019). Abadi et al. (2016) proposed a differentially private version of stochastic gradient descent (SGD), called DP-SGD, to enable differentially private training of generic ML models. Their analysis has further been refined (Mironov, 2017; Mironov et al., 2019) and has been shown experimentally to be tight (Nasr et al., 2021).

## 3 DIFFICULTY CALIBRATION

As depicted in Figure 1, prior works on score-based membership inference attack are very unreliable for separating *easy-to-predict non-members* from *hard-to-predict members* since both can attain a high membership score. We argue that a simple modification to the score, which we call *difficulty calibration*, can drastically improve the attack's reliability. This approach has been applied to the loss score for high-precision attack against well-generalized models (Long et al., 2018).

Let $s(h, (\boldsymbol{x}, y))$ be the membership score, where higher score indicates a stronger signal that the sample $(\boldsymbol{x}, y)$ is a member. Instead of computing the membership score only on the target model $h$, we sample multiple "typical" models trained on the same data distribution as $h$ and evaluate the membership score on these models. Doing so calibrates $s(h, (\boldsymbol{x}, y))$ to the difficulty of the sample. For instance, if $(\boldsymbol{x}, y)$ is an easy-to-predict non-member, then $s(h, (\boldsymbol{x}, y))$ is high but typical models $g$ would also perform well on $(\boldsymbol{x}, y)$, hence $s(g, (\boldsymbol{x}, y))$ would also be high. The small gap between $s(h, (\boldsymbol{x}, y))$ and $s(g, (\boldsymbol{x}, y))$ suggests that $(\boldsymbol{x}, y)$ is likely a non-member.

Formally, let $\mathcal{D}_{\text{shadow}}$ be a shadow dataset drawn from the same data distribution as the training set of $h$, and let $\mathcal{A}$ be a randomized training algorithm that samples from a distribution over models trained on $\mathcal{D}_{\text{shadow}}$. We define the calibrated score as:

$$s^{\text{cal}}(h, (\boldsymbol{x}, y)) = s(h, (\boldsymbol{x}, y)) - \mathbb{E}_{g \leftarrow \mathcal{A}(\mathcal{D}_{\text{shadow}})}[s(g, (\boldsymbol{x}, y))], \tag{5}$$

where the expectation is approximated by sampling one or more models from $\mathcal{A}(\mathcal{D}_{\text{shadow}})$. The membership inference attack proceeds by thresholding on the score $s^{\text{cal}}(h, (\boldsymbol{x}, y))$.

**Efficient difficulty calibration via forgetting.** Faithfully executing the difficulty calibration algorithm in Equation 5 requires training multiple models $g$ using the randomized training algorithm $\mathcal{A}$. This can be prohibitively expensive if the model is large, which is the case with modern neural networks such as large-scale transformers (Brown et al., 2020).

An alternative, more efficient approach if given white-box access to the target model $h$ is to warm-start training on $h$. In fact, this approach establishes an explicit connection between membership inference attacks and *catastrophic forgetting* in neural networks (Goodfellow et al., 2013; Kirkpatrick et al., 2017). Toneva et al. (2018) observed that when a trained model resumes training on a separate dataset drawn from the same data distribution, it is very likely to "forget" about its original training samples. The training examples that are most likely forgotten are ones with abnormal or distinctive features, and this set of examples is stable across different training runs. We leverage this phenomenon to define a more efficient variant of difficulty calibration.

**Connection to posterior inference.** Under a simplifying assumption of the distribution $\mathcal{A}(\mathcal{D})$ induced by the training algorithm, it is possible to derive difficulty calibration as an approximation to an optimal white-box attack. Sablayrolles et al. (2019) proposed a formal analysis of membership inference, where they assumed that the density function of $\mathcal{A}(\mathcal{D})$ has the form:

$$p(h; \mathcal{D}) \propto e^{-\sum_{(\boldsymbol{x},y) \in \mathcal{D}} \ell(h(\boldsymbol{x}), y)}, \tag{6}$$

which can be formally derived for optimization using SGD under certain conditions (Sato & Nakagawa, 2014). The implied threat model in differential privacy assumes that the adversary has knowledge that the training dataset is either $\mathcal{D}$ or $\mathcal{D}' = \mathcal{D} \cup \{(\boldsymbol{x}, y)\}$ for some inference target $(\boldsymbol{x}, y)$, and $\mathcal{D}$ is known to the adversary. The Bayes-optimal attack strategy under this setting can be approached by thresholding on the calibrated score in Equation 5, where $s = s_{\text{loss}}$ and $\mathcal{D} = \mathcal{D}_{\text{shadow}}$ (Sablayrolles et al., 2019). This analysis can be extended to any scoring function $s(h, (\boldsymbol{x}, y))$, such as gradient norm or confidence (Equations 2,3).

## 4 EXPERIMENTS

To demonstrate the effect of difficulty calibration, we perform a comprehensive evaluation of several score-based attacks on standard benchmark datasets.[1] We evaluate both variants of difficulty calibration: (i) calibration by training reference models from scratch, and (ii) calibration via forgetting.

### 4.1 EXPERIMENTAL SETUP

**Datasets.** We perform experiments on several benchmark classification datasets: *German Credit*, *Hepatitis* and *Adult* datasets from the UCI Machine Learning Repository (Dua & Graff, 2017), *MNIST* (LeCun et al., 1998), *CIFAR10/100* (Krizhevsky et al., 2009), and *ImageNet* (Deng et al., 2009). The datasets vary in size from 155 to 1,281,167 points and represent both image and non-image classification tasks.

**Attack setup.** We split the data into two sets: a *private* set, known only to the trainer, and a *public* set, which is used for training reference models and selecting the decision threshold $\tau$. The trainer trains their model $h$ on half of the private set, keeping the other half as non-members. At evaluation time, a sample $(\boldsymbol{x}, y)$ is selected from the private set. The attacker computes a membership score $s(h, (\boldsymbol{x}, y))$ and predicts $(\boldsymbol{x}, y)$ as a member if $s(h, (\boldsymbol{x}, y)) > \tau$. For ImageNet, we sample 10,000 member and non-member samples from the member and non-member sets, while for all other datasets we use the full private set for evaluation. With such a setup, the accuracy of a random adversary is 50%, as is the precision at all levels of recall.

We repeat this attack under the same setup $T$ times, each time performing a different random split into the private/public and member/non-member sets. We set $T = 5$ for all datasets except for

---

[1]An implementation of these attacks is available at `https://github.com/facebookresearch/calibration_membership`.

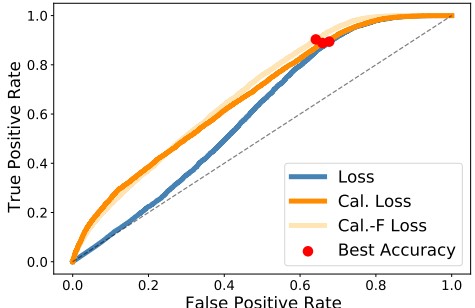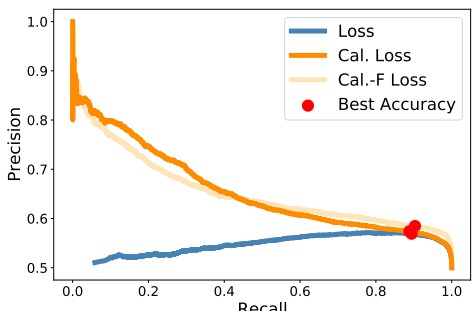

Figure 2: Left: ROC (left) and precision-recall (right) curves of calibrated/uncalibrated loss score attacks on CIFAR10. The threshold $\tau$ that optimizes accuracy is shown as a red dot. Calibration yields a higher TPR for the same value of FPR, or equivalently a higher precision at low levels of recall. The precision-recall trade-off surfaces very different behaviors for two methods that have otherwise very similar accuracy.

ImageNet, where $T = 3$. Thus, in all our results, the reported averages and standard deviations reflect randomness in both model optimization and in the data splits.

**Model Architectures.** The target model for the German Credit, Adult and Hepatitis datasets is a multi-layer perceptron (MLP) with one hidden layer and ReLU activation followed by a softmax layer. For a dataset with $m$ features, the hidden layer has $2m$ hidden units. For MNIST, we followed the architecture used in Leino & Fredrikson (2020), which is a small convolutional network with two convolutional layers with 20 and 50 output channels respectively, and a kernel size of 5, followed by max pooling layers. The classification head consists of one fully connected layer with 500 hidden units and ReLU activation. Dropout at rate 0.25 is applied after the max pooling layers and at rate 0.5 after the fully connected layer. For CIFAR10 and CIFAR100, we used a CNN architecture with 4 convolutional and average pooling layers and ReLU activations using 32, 64, 64 and 128 output channels respectively with kernel size 3. For ImageNet, we used the ResNet18 (He et al., 2016) model architecture.

**Model training.** The target models are trained for between 50 and 200 epochs, with batch sizes varying from 4 (for very small datasets) to 1024. For optimization, we use SGD with a learning rate of 0.1, Nesterov momentum of 0.9 and a cosine learning rate schedule for the CIFAR10/100 and ImageNet datasets. Smaller datasets such as the German Credit dataset also used weight decay of $1 \times 10^{-4}$. Differentially private training is done using the Opacus (Yousefpour et al., 2021) library. Table 1 presents the target model accuracy statistics.

Reference models for difficulty calibration are trained using the same hyperparameters as the target model. For most experiments, we train a single reference model for calibration, and show the effect of the number of reference models in subsection 4.3. For calibration via forgetting, reference models were tuned over a set of learning rates and batch sizes to ensure that their training and test accuracies were similar to the target model for their largest number of training epochs. For both calibration methods, the reference models were trained until their validation accuracy reached its maximum value, and stopped before their training accuracy converged to

| Task | Train Acc. | Test Acc. |
|---|---|---|
| Credit | 0.921 | 0.751 |
| Hep. | 0.998 | 0.871 |
| Adult | 0.904 | 0.842 |
| MNIST | 0.990 | 0.989 |
| CIFAR10 | 0.946 | 0.740 |
| CIFAR100 | 0.938 | 0.373 |
| ImageNet | 0.773 | 0.637 |

Table 1: Average train and test accuracy statistics for the target models.

its maximum. Due to its warm start, calibration-via-forgetting generally used fewer training epochs for reference models.

**Threshold selection.** When computing AUC and precision/recall metrics, we sweep over a range of values for the threshold $\tau$ and measure the resulting attack's FPR/TPR and precision/recall trade-offs. To find a threshold for optimal accuracy, we first split the public set of examples in half again, and treat one half as members, with the rest as non-members. We can then choose the best threshold

| Dataset | Gap Atk. | Loss | | | GN | | | Confidence | | |
|---|---|---|---|---|---|---|---|---|---|---|
| | | Orig | Cal. | Cal.-F | Orig | Cal. | Cal.-F | Orig | Cal. | Cal.-F |
| Credit | 0.549 | 0.542 | **0.648** | 0.594 | 0.513 | 0.579 | 0.559 | 0.501 | 0.553 | 0.543 |
| Hep. | 0.535 | 0.514 | 0.558 | **0.559** | 0.515 | 0.553 | 0.548 | 0.513 | 0.525 | 0.533 |
| Adult | 0.514 | 0.516 | **0.554** | 0.529 | 0.510 | 0.523 | 0.515 | 0.507 | 0.510 | 0.512 |
| MNIST | 0.506 | 0.505 | **0.519** | 0.509 | 0.504 | 0.514 | 0.509 | 0.503 | 0.510 | 0.505 |
| CIFAR10 | 0.663 | 0.676 | 0.731 | 0.708 | 0.678 | **0.787** | 0.762 | 0.629 | 0.635 | 0.631 |
| CIFAR100 | 0.854 | 0.911 | 0.903 | 0.886 | 0.912 | **0.924** | 0.912 | 0.852 | 0.707 | 0.760 |
| ImageNet | 0.536 | 0.547 | **0.557** | 0.551 | 0.540 | 0.511 | 0.520 | 0.544 | 0.515 | 0.512 |
| CIFAR10 (aug.) | 0.603 | 0.603 | 0.690 | 0.674 | 0.603 | 0.666 | **0.707** | 0.560 | 0.568 | 0.575 |
| CIFAR100 (aug.) | 0.784 | 0.812 | 0.844 | 0.831 | 0.811 | **0.856** | 0.843 | 0.679 | 0.672 | 0.647 |

Table 2: AUC metric for score-based membership attacks before and after difficulty calibration. The standard deviation is up to 0.024 for Credit, up to 0.034 for Hep., up to 0.003 for MNIST and up to 0.015 for all other datasets. Calibration (Cal.) consistently improves the AUC of attacks by a significant margin, while calibration-via-forgetting (Cal.-F) sacrifices a modest amount of improvement for better efficiency.

| Dataset | Gap Atk. | Loss | | | GN | | | Confidence | | |
|---|---|---|---|---|---|---|---|---|---|---|
| | | Orig | Cal. | Cal.-F | Orig | Cal. | Cal.-F | Orig | Cal. | Cal.-F |
| Credit | 0.589 | 0.617 | **0.618** | 0.577 | 0.569 | 0.573 | 0.567 | 0.557 | 0.554 | 0.555 |
| Hep. | 0.561 | 0.574 | 0.575 | 0.591 | 0.574 | **0.593** | 0.585 | 0.574 | 0.577 | 0.576 |
| Adult | 0.534 | **0.536** | 0.534 | 0.522 | 0.518 | 0.518 | 0.515 | 0.512 | 0.511 | 0.511 |
| MNIST | 0.506 | 0.508 | **0.513** | 0.509 | 0.508 | 0.511 | 0.508 | 0.507 | 0.508 | 0.506 |
| CIFAR10 | 0.664 | 0.712 | 0.657 | 0.662 | 0.719 | **0.720** | 0.702 | 0.642 | 0.623 | 0.622 |
| CIFAR100 | 0.854 | 0.911 | 0.829 | 0.862 | **0.915** | 0.876 | 0.898 | 0.820 | 0.731 | 0.711 |
| ImageNet | 0.536 | 0.542 | 0.545 | 0.542 | 0.538 | 0.518 | 0.521 | 0.540 | 0.541 | **0.553** |
| CIFAR10 (aug.) | 0.602 | 0.609 | 0.626 | 0.625 | 0.610 | 0.610 | **0.651** | 0.562 | 0.542 | 0.558 |
| CIFAR100 (aug.) | 0.784 | 0.785 | 0.765 | 0.775 | 0.788 | **0.793** | 0.789 | 0.647 | 0.641 | 0.621 |

Table 3: Attack accuracy for score-based membership attacks before and after difficulty calibration. The standard deviation is up to 0.018 for Credit, up to 0.026 for Hep., up to 0.002 for MNIST and up to 0.005 for all other datasets. In almost all settings, calibrated attacks are on-par with or better than their uncalibrated version in terms of accuracy.

$\tau$ based on membership inference accuracy for this simulated setup. Frequently, the optimal threshold for calibrated attacks was a value only slightly greater than 0 (e.g., 0.0001), which can be used as a default threshold if necessary.

## 4.2 MAIN RESULTS

We first evaluate attacks that rely on continuous-valued output from the target model: loss score (Equation 1), gradient norm (GN) score (Equation 2), and confidence score (Equation 3). We also consider the gap attack (Yeom et al., 2018) as a baseline to measure progress against. See Appendix C for additional experiments using multiple calibration models and other membership scores.

**AUC and accuracy.** We compute two primary metrics: area under ROC curve (AUC) for evaluating the FPR/TPR trade-off, and attack accuracy. Table 2 shows the computed AUC for various attacks before and after difficulty calibration. Overall, calibrated attacks often perform drastically better than their uncalibrated counterparts, and in some instances the performance increase can exceed 0.10, such as on the German Credit and CIFAR10 datasets. Table 3 shows the corresponding attack accuracy numbers. Evidently, calibration does not reduce accuracy in most cases, and sometimes brings a modest improvement.

We observe that the confidence score attacks under-perform other methods due to the model predicting non-member points incorrectly with high confidence. For this reason, calibrated confidence did not improve the attack performance on many datasets, as training points that were correctly classified on the target model but were incorrectly classified on the reference model became indistinguishable.

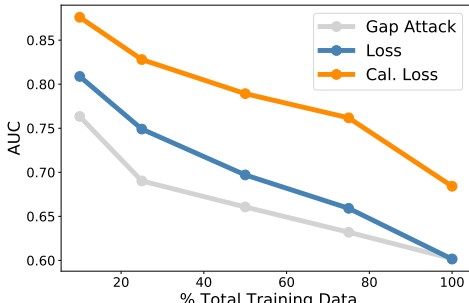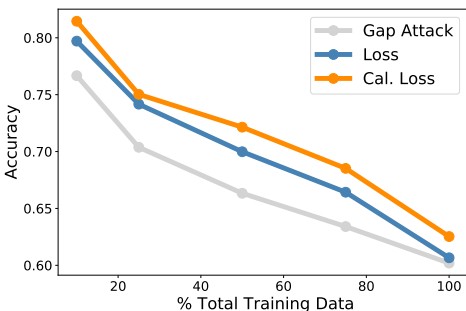

Figure 3: AUC (left) and accuracy (right) of the gap attack and calibrated/uncalibrated loss score attacks against target model trained on varying training set sizes on CIFAR10. Both AUC and accuracy increase as the training dataset size decreases due to more severe overfitting of the target model. Difficulty calibration can effectively leverage this to improve the attack's AUC and accuracy across all training set sizes.

**Effect of data augmentation.** The last two rows in Tables 2 and 3 show various attacks' performance on the CIFAR10/100 target model trained *with* data augmentation. As expected, attacks are less effective on models with data augmentation due to reduced overfitting. Nevertheless, calibration still improves both the AUC and accuracy of loss and gradient norm based attacks most of the time.

**ROC and precision-recall curves.** Figure 2 shows the ROC and precision-recall curves for the loss score attack on CIFAR10, which demonstrates the effect of difficulty calibration in closer detail. In the left plot, at low FPR values, the uncalibrated attack has close to zero advantage in TPR, behaving essentially the same as random guessing. In stark contrast, calibrated attacks exhibit a large difference in the FPR and TPR values. Furthermore, as shown in the right plot, calibrated attacks can detect a group of member points with very high precision at low recall values, which is not possible for the uncalibrated counterpart. This pattern is also seen in the membership scores of calibrated and uncalibrated attacks shown in Figure 1.

We highlight another interesting phenomenon by computing the threshold $\tau$ that achieves optimal accuracy for each attack, shown by the red dot in Figure 2. For all attacks, this threshold lies towards the tail of the ROC and precision-recall curves, where the attack achieves high FPR and high recall but low precision. This observation concurs with prior work, which shows that existing attacks do not predict membership reliably when optimizing for accuracy (Rezaei & Liu, 2021).

**Label-only attacks.** When the target model returns only a discrete label rather than a continuous-valued output, membership inference can still be performed using label-only attacks. We show that difficulty calibration also improves such attacks. Table 4 reports the performance of calibrated and uncalibrated label-only attacks using the HopSkipJump boundary distance method (Choquette-Choo et al., 2021). We excluded the MNIST and ImageNet datasets as the HopSkipJump attack required too many queries to successfully change the label. Similar to the results in Tables 2 and 3, label-only attacks also benefit from difficulty calibration with increased AUC and improved or similar accuracy. However, the performance increase is noticeably smaller for CIFAR10/100 compared to other score-based attacks.

## 4.3 ABLATION STUDIES

**Training set size.** Since vulnerability to membership inference attack is linked to overfitting (Yeom et al., 2018), the training set size of the target model is a crucial factor in attack performance. Indeed, Figure 3 shows that for the gap attack and calibrated/uncalibrated loss score attacks, both AUC and accuracy increase when the training set size is reduced. More importantly, difficulty calibration can effectively leverage overfitting to improve the attack's performance across the entire range of training set sizes.

**Ratio of member to non-member samples.** The experimental setup so far uses an equal number of member and non-member samples. In real world scenarios, the majority of samples are likely

| Privacy | AUC | | | Accuracy | | |
|---|---|---|---|---|---|---|
| | Orig. | Cal. | Cal.-F | Orig. | Cal. | Cal.-F |
| Credit | 0.552 | **0.640** | 0.599 | 0.616 | 0.620 | **0.638** |
| Hep. | 0.547 | 0.585 | **0.594** | **0.596** | 0.588 | 0.591 |
| Adult | 0.512 | **0.541** | 0.537 | 0.533 | **0.549** | 0.539 |
| CIFAR10 | 0.660 | **0.686** | 0.620 | 0.665 | **0.666** | 0.623 |
| CIFAR100 | 0.849 | 0.841 | **0.856** | 0.847 | 0.841 | 0.844 |

Table 4: Average accuracy and AUC of the HopSkipJump label-only attack using $\min(n, 1000)$ datapoints for each of the training and heldout sets, where $n$ is their original datasize. CIFAR models were trained without data augmentation.

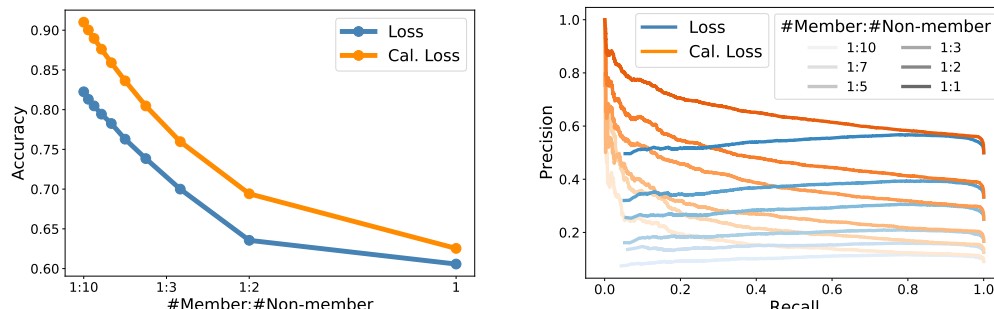

Figure 4: Accuracy (left) and precision-recall curve (right) of the loss score attack when changing the member to non-member ratio. At low ratios, the calibrated attack can still identify members with high precision, whereas the uncalibrated attack is unable to do so despite high accuracy.

non-members, which drastically affects the accuracy and precision-recall trade-off of membership inference attacks. In the following experiment, we simulate this scenario by subsampling the member set for evaluation to decrease the ratio of member to non-member samples.

Figure 4 shows the accuracy and precision-recall curves for different member to non-member ratios. In the left plot, as the number of non-member samples increases, attack accuracy also drastically increases. This is due to the threshold $\tau$ for optimal accuracy favoring high recall and low precision (see Figure 2), effectively performing "non-membership inference". In contrast, the right plot shows that precision-recall trade-off worsens when the number of non-member samples increases, which reflects the fact that attacks are less reliable in the real world when most samples are non-members. Nevertheless, for the calibrated attack, it is possible to achieve a high precision at low values of recall even for low member to non-member ratios.

**Number of reference models.** Results reported in Section 4.2 use a single reference model for difficulty calibration. Figure 5 shows the effect of increasing the number of reference models on AUC and attack accuracy. To train each reference model, we subsample a different subset of the public dataset without replacement to create the shadow dataset. We then apply calibration-via-forgetting to calibrate the loss score, while varying the shadow dataset size. Increasing the number of reference models and increasing the shadow dataset size strictly improve both AUC and accuracy. The difference between 1 and 10 reference models is up to 0.05 for AUC and 0.03 for accuracy.

**Differential privacy** is by far the most common defense against membership inference attacks, and provably limits the accuracy of these attacks (Yeom et al., 2018) when $\epsilon$ is small. With a higher value of $\epsilon$, DP still empirically protects against uncalibrated membership inference attacks *on average*; however, calibrated attacks provide a significant edge against outliers for high values of the privacy parameter $\epsilon$.

Figure 6 shows the ROC and precision-recall curves of calibrated/uncalibrated loss score attacks on CIFAR10 models trained with DP-SGD. As expected, DP clearly reduces the effectiveness of attacks as indicated by the gap between attacks against models with finite $\epsilon$ (blue and orange lines) and at-

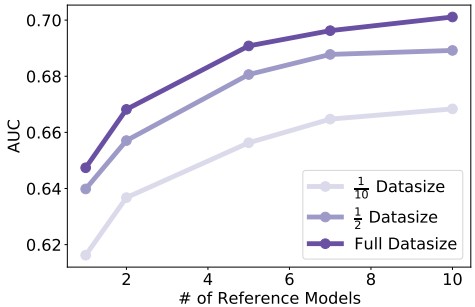 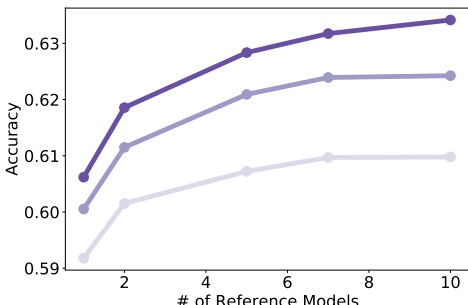

Figure 5: AUC (left) and attack accuracy (right) for the calibrated loss attack with varying number of reference models on CIFAR-10. Using more reference models and training reference models with a higher shadow dataset size strictly improve performance at a cost of more computation.

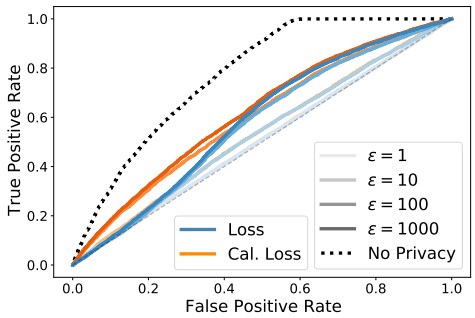 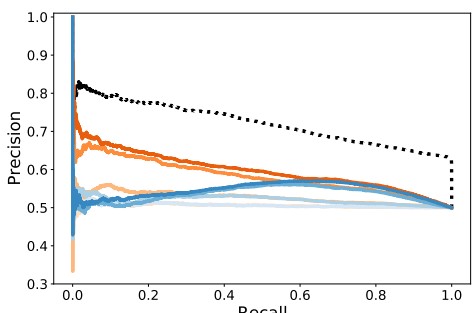

Figure 6: ROC (left) and precision-recall (right) curves for calibrated/uncalibrated loss score attack on CIFAR10 against differentially private models. For large values of DP parameter $\epsilon$, the calibrated attack can still successfully infer member samples with high precision at low recall.

tacks against an undefended model (dotted line). However, for $\epsilon = 100$ and $1000$, calibrated attacks can still attain a non-trivial FPR-TPR and precision-recall trade-off. Only when $\epsilon$ approaches 1 do the AUC and accuracy approach that of uncalibrated attacks, and both converge to the effectiveness of random guessing.

A qualitative inspection of member images from CIFAR10 that are successfully inferred at a high precision of $> 0.9$ for the DP model at $\epsilon = 1000$, but are not identified for $\epsilon \leq 10$, showed that vulnerable images were often atypical samples from their respective classes. They did not appear to be extreme outliers. A further manual inspection of examples did not suggest any general pattern for the most vulnerable points.

## 5    CONCLUSION

We showed that difficulty calibration is a powerful post-processing technique for improving existing membership inference attacks. Most score-based attacks can greatly benefit from calibration, making the attack much more reliable in low false positive rate and high precision regimes. Our study also showed that prior work that focused on evaluating attack accuracy do not reveal the full picture, and detailed analysis using the ROC and precision-recall curves is important for designing reliable membership inference attacks.

When given white-box access to the target model, calibration via forgetting can be an efficient alternative to training reference models from scratch. This alternative method also serves as an explicit connection between membership inference and the well-studied phenomenon of catastrophic forgetting in neural networks. We hope that future work can further explore this connection and leverage new findings to improve the state-of-the-art in membership inference attacks.

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

# A   CALIBRATED GAP ATTACK

In this section, we explore the effects of using difficulty calibration on the *gap attack* Yeom et al. (2018). This analysis provides an intuitive explanation as to why calibration improves the AUC of many attacks while retaining similar accuracy statistics in comparison to their uncalibrated counterparts.

| calib \ target | correct | incorrect |
|---|---|---|
| correct | $p_{\text{test}}$ | $0$ |
| incorrect | $p_{\text{train}} - p_{\text{test}}$ | $1 - p_{\text{train}}$ |

| calib \ target | correct | incorrect |
|---|---|---|
| correct | $p_{\text{test}}$ | $0$ |
| incorrect | $0$ | $1 - p_{\text{test}}$ |

Table 5: Performance of the calibrated and target models on the train (left) and test (right) sets.

Let us assume that we have two models, *target* and *calib*. The calibrated gap attack predicts that a sample comes from the training set if the target model predicts it correctly and the calibration model predicts it incorrectly.

**Simplified analysis.**   We simplify the analysis by making two assumptions:

- on the train set, if the target model is wrong, the calibrated model is wrong too
- on the test set, the models are either both right or both wrong about a sample

Table 5 shows that we will predict all of the test set elements correctly, and a portion $p_{\text{train}} - p_{\text{test}}$ of the train set correctly. The accuracy of such a method will thus be

$$\frac{1}{2}\left(1 + p_{\text{train}} - p_{\text{test}}\right). \tag{7}$$

| calib \ target | correct | incorrect | calib \ target | correct | incorrect |
|---|---|---|---|---|---|
| correct | $p_{\text{test}} - \epsilon_1$ | $\epsilon_1$ | correct | $p_{\text{test}} - \epsilon_2$ | $\epsilon_2$ |
| incorrect | $p_{\text{train}} - p_{\text{test}} + \epsilon_1$ | $1 - p_{\text{train}} - \epsilon_1$ | incorrect | $\epsilon_2$ | $1 - p_{\text{test}} - \epsilon_2$ |

Table 6: Performance of the calibrated and target models on the train (left) and test (right) sets.

**Full analysis.**   Now let us get rid of this assumption. In order to fill Table 6, we have 3 equations: all probabilities should sum to one, and the rows and columns where the model is correct should sum to $p_{\text{train}}$ (respectively $p_{\text{test}}$). Given that we have four unknowns, this leaves only one undetermined quantity for each table, $\epsilon_1$ and $\epsilon_2$. The accuracy of the calibrated gap attack is thus

$$\frac{1}{2}\left(1 - \epsilon_2 + p_{\text{train}} - p_{\text{test}} + \epsilon_1\right). \tag{8}$$

Based on experiments using the calibrated gap attack on CIFAR10 and CIFAR100 with and without data augmentation, we observed that $\epsilon_1$ is often very small, with values ranging from $0\% - 2\%$. On the other hand, $\epsilon_2$ is often larger at around $8\% - 12\%$.

**Accuracy:** As implied by Equation 8, the accuracy of the calibrated gap attack is then lower than the uncalibrated gap attack as $\epsilon_2$ exceeds $\epsilon_1$. However, note that the accuracy of the uncalibrated gap attack can be trivially recovered in this context by simply predicting that all points predicted correctly by the target model are members, as opposed to only those predicted correctly by the target model and incorrectly by the calibration model.

**AUC:** On the other hand, the calibrated gap attack improves the precision-recall trade-off of the gap attack. As shown in Figures 7 and 8, calibration makes it possible to identify a subset of member points with a significantly lower false positive rate. For this reason, the AUC of the gap attack on CIFAR10 improved from an average of 0.66 to 0.71.

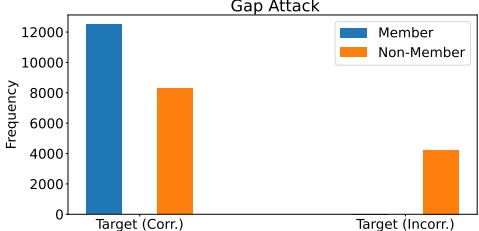 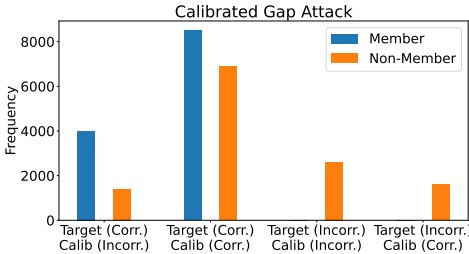

Figure 7: Gap Attack vs. Calibrated Gap Attack score distributions. Target (Corr.) indicates that the predicted class for the datapoint was correct using the target model. Calib (Incorr.) indicates that the predicted class was incorrectly predicted on the calibration model. Note that the two left-most categories in the calibrated plot correspond to a division of the left hand category in the original gap attack plot. The same relationship exists between the right-most categories. The left-most category in the calibrated gap attack shows the improvement in TPR that can be attained via calibration, resulting in the increased AUC seen in Figure 8.

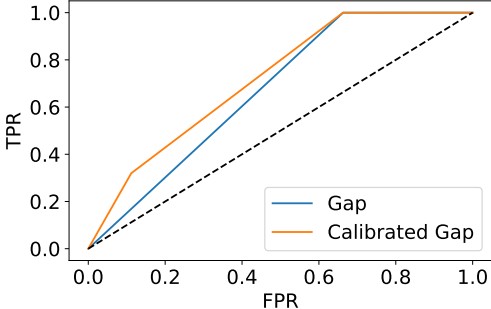

Figure 8: The ROC curves for the gap attack and calibrated gap attack. Note that the area under the ROC curve (AUC) is higher for the calibrated attack due to the increased TPR for low FPR.

## B    GENERALIZATION AND MEMBERSHIP INFERENCE

Calibrated membership inference looks at the quantity $\ell(\theta_0, z) - \ell(\theta, z)$ where $\theta$ is the target model, $\theta_0$ the calibration model and $z$ a sample included in the training set of $\theta$. Generalization, on the other hand, is concerned with $\ell(\theta, z') - \ell(\theta, z)$ where $z'$ is a sample that does not belong to the training set of $\theta$. We argue that $\ell(\theta_0, z)$ and $\ell(\theta, z')$ are different quantities but they both represent the loss that is incurred on a sample outside of the training set and should thus behave similarly statistically.

## C    ADDITIONAL EXPERIMENTS

Tables 7 and 8 demonstrate the impact of using multiple calibration models instead of a single calibration model. Tables 9 and 10 show the effects of difficulty calibration on other membership scores not included in the main body of this paper. Specifically, we considered the entropy and modified entropy scores proposed by Song & Mittal (2021) as well as the Merlin and Morgan attacks proposed by Jayaraman et al. (2021). As suggested by Song & Mittal (2021), Figure 9 shows that their entropy metrics are closely related to the confidence and loss scores respectively. This relationship explains why the AUC and accuracy results show the same behaviours as the confidence and loss scores under calibration.

**Merlin and Morgan:** The Merlin and Morgan attacks target high positive predictive value (PPV) as opposed to attack accuracy. For this reason, their accuracy and AUC results are lower than that of other attacks. In these attacks, the member and non-member distributions are often indistinguishable except for the small number of points identified with very high PPV. In particular, the Morgan attack outputs binary values chosen to maximize PPV. When successful this results in a very skewed

distribution identifying a few datapoints with PPV close to 1 (and value 1) and assigning all other points value 0. This results in accuracy and AUC results close to that of random guessing. Table 11 reports the PPV of uncalibrated and calibrated versions of the Merlin and Morgan attacks, in many cases calibration improves the maximum achieved PPV.

| Dataset | Gap Atk. | Loss | | | GN | | | Confidence | | |
|---|---|---|---|---|---|---|---|---|---|---|
| | | Orig | Cal. | Cal.-F | Orig | Cal. | Cal.-F | Orig | Cal. | Cal.-F |
| Credit | 0.589 | 0.542 | **0.668** | 0.650 | 0.513 | 0.597 | 0.596 | 0.501 | 0.569 | 0.550 |
| Hep. | 0.561 | 0.514 | 0.562 | 0.534 | 0.515 | 0.557 | 0.568 | 0.513 | **0.582** | 0.573 |
| Adult | 0.534 | 0.516 | **0.559** | 0.539 | 0.510 | 0.527 | 0.520 | 0.507 | 0.518 | 0.513 |
| MNIST | 0.506 | 0.505 | **0.527** | 0.510 | 0.504 | 0.525 | 0.517 | 0.503 | 0.517 | 0.509 |
| CIFAR10 | 0.663 | 0.676 | 0.739 | 0.696 | 0.678 | 0.796 | **0.759** | 0.629 | 0.658 | 0.646 |
| CIFAR100 | 0.854 | 0.911 | 0.914 | 0.916 | 0.912 | **0.933** | 0.918 | 0.852 | 0.721 | 0.746 |
| CIFAR10 (aug.) | 0.603 | 0.603 | 0.704 | 0.693 | 0.603 | **0.746** | 0.725 | 0.560 | 0.581 | 0.578 |
| CIFAR100 (aug.) | 0.784 | 0.812 | 0.878 | 0.873 | 0.811 | **0.891** | 0.877 | 0.679 | 0.665 | 0.635 |

Table 7: AUC metric for score-based membership attacks before and after difficulty calibration with 10 calibration models. Calibration (Cal.) consistently improves the AUC of attacks by a significant margin, while calibration-via-forgetting (Cal.-F) sacrifices a modest amount of improvement for better efficiency.

| Dataset | Gap Atk. | Loss | | | GN | | | Conf | | |
|---|---|---|---|---|---|---|---|---|---|---|
| | | Orig | Cal. | Cal.-F | Orig | Cal. | Cal.-F | Orig | Cal. | Cal.-F |
| Credit | 0.589 | 0.617 | **0.624** | 0.605 | 0.569 | 0.586 | 0.584 | 0.557 | 0.563 | 0.553 |
| Hep. | 0.561 | 0.574 | 0.587 | 0.574 | 0.574 | 0.590 | 0.593 | 0.574 | 0.601 | **0.594** |
| Adult | 0.534 | 0.536 | **0.537** | 0.525 | 0.518 | 0.520 | 0.516 | 0.512 | 0.514 | 0.511 |
| MNIST | 0.506 | 0.508 | **0.517** | 0.510 | 0.508 | 0.516 | 0.512 | 0.507 | 0.513 | 0.508 |
| CIFAR10 | 0.664 | 0.712 | 0.657 | 0.663 | **0.719** | 0.707 | 0.703 | 0.642 | 0.614 | 0.623 |
| CIFAR100 | 0.854 | 0.911 | 0.832 | 0.846 | **0.915** | 0.885 | 0.896 | 0.820 | 0.675 | 0.691 |
| CIFAR10 (aug.) | 0.602 | 0.609 | 0.631 | 0.639 | 0.610 | **0.668** | 0.657 | 0.562 | 0.567 | 0.564 |
| CIFAR100 (aug.) | 0.784 | 0.785 | 0.791 | 0.801 | 0.788 | **0.831** | 0.822 | 0.647 | 0.616 | 0.606 |

Table 8: Attack accuracy for score-based membership attacks before and after difficulty calibration with 10 calibration models. In almost all settings, calibrated attacks are on-par with or better than their uncalibrated version in terms of accuracy.

| Dataset | Gap | Entropy | | | Modified Entropy | | | Merlin | | | Morgan | | |
|---|---|---|---|---|---|---|---|---|---|---|---|---|---|
| | | Orig. | Cal. | Cal.-F | Orig. | Cal. | Cal.-F | Orig. | Cal. | Cal.-F | Orig. | Cal. | Cal.-F |
| Credit | 0.588 | 0.525 | 0.591 | 0.597 | 0.562 | **0.637** | 0.621 | 0.514 | 0.523 | 0.527 | 0.505 | 0.502 | 0.508 |
| Hep. | 0.544 | 0.516 | 0.520 | 0.534 | 0.534 | 0.581 | **0.587** | 0.472 | 0.476 | 0.484 | 0.524 | 0.544 | 0.526 |
| Adult | 0.514 | 0.509 | 0.512 | 0.513 | 0.519 | **0.557** | 0.530 | 0.500 | 0.502 | 0.507 | 0.500 | 0.500 | 0.500 |
| MNIST | 0.506 | 0.507 | 0.512 | 0.510 | 0.510 | 0.512 | **0.513** | 0.497 | 0.503 | 0.501 | 0.503 | 0.502 | 0.503 |
| CIFAR10 | 0.662 | 0.627 | 0.634 | 0.635 | 0.675 | **0.730** | 0.710 | 0.506 | 0.504 | 0.504 | 0.500 | 0.500 | 0.500 |
| CIFAR100 | 0.858 | 0.849 | 0.701 | 0.750 | 0.910 | **0.912** | 0.902 | 0.373 | 0.530 | 0.510 | 0.500 | 0.500 | 0.500 |
| CIFAR10 (aug.) | 0.602 | 0.563 | 0.598 | 0.581 | 0.603 | **0.691** | 0.688 | 0.500 | 0.516 | 0.514 | 0.555 | 0.500 | 0.500 |
| CIFAR100 (aug.) | 0.785 | 0.664 | 0.700 | 0.706 | 0.798 | 0.851 | **0.868** | 0.495 | 0.507 | 0.509 | 0.644 | 0.500 | 0.500 |

Table 9: AUC results. As suggested by Song & Mittal (2021), Figure 9 shows that the entropy and modified entropy metrics are closely related to the confidence and loss scores respectively. This relationship explains why the AUC results are similar to those of the confidence and loss scores under calibration. The Merlin and Morgan attacks were proposed to maximize positive predictive value (PPV) instead of accuracy, which is reflected in their results. Note that there was a large standard deviation for the Hepatitis dataset using the Merlin and Morgan attacks of up to 0.09 making the results difficult to compare. For the Morgan attack, a value close to 0.500 results from the attack identifying a small number of points with $FPR = 0$ and achieving the attack's objective of maximizing PPV. Higher values indicate when this was not possible, and the intended behaviour was then improved by calibration due to the creation of a high precision region e.g. see CIFAR100(aug) and Table 11.

| Dataset | Gap | Entropy | | | Modified Entropy | | | Merlin | | | Morgan | | |
|---|---|---|---|---|---|---|---|---|---|---|---|---|---|
| | | Orig. | Cal. | Cal.-F | Orig. | Cal. | Cal.-F | Orig. | Cal. | Cal.-F | Orig. | Cal. | Cal.-F |
| Credit | 0.588 | 0.525 | 0.591 | 0.597 | 0.622 | **0.633** | 0.629 | 0.537 | 0.537 | 0.543 | 0.505 | 0.502 | 0.508 |
| Hep. | 0.544 | 0.516 | 0.520 | 0.534 | **0.597** | **0.597** | 0.591 | 0.544 | 0.571 | 0.558 | 0.524 | 0.544 | 0.526 |
| Adult | 0.514 | 0.509 | 0.512 | 0.513 | **0.537** | 0.535 | 0.533 | 0.505 | 0.505 | 0.507 | 0.500 | 0.500 | 0.500 |
| MNIST | 0.506 | 0.507 | **0.512** | 0.510 | 0.510 | 0.509 | 0.509 | 0.502 | 0.505 | 0.504 | 0.503 | 0.502 | 0.503 |
| CIFAR10 | 0.662 | 0.627 | 0.634 | 0.635 | **0.710** | 0.661 | 0.664 | 0.515 | 0.504 | 0.511 | 0.500 | 0.500 | 0.500 |
| CIFAR100 | 0.858 | 0.849 | 0.701 | 0.750 | **0.910** | 0.850 | 0.839 | 0.501 | 0.532 | 0.525 | 0.500 | 0.500 | 0.500 |
| CIFAR10 (aug.) | 0.602 | 0.563 | 0.598 | 0.581 | 0.608 | 0.632 | **0.634** | 0.501 | 0.516 | 0.513 | 0.555 | 0.500 | 0.500 |
| CIFAR100 (aug.) | 0.785 | 0.664 | 0.700 | 0.706 | 0.771 | **0.801** | 0.793 | 0.502 | 0.522 | 0.524 | 0.644 | 0.500 | 0.500 |

Table 10: Accuracy results. As suggested by Song & Mittal (2021), Figure 9 shows that the entropy and modified entropy metrics are closely related to the confidence and loss scores respectively. This relationship explains why the accuracy results are similar to those of the confidence and loss scores under calibration. The Merlin and Morgan attacks were proposed to maximize positive predictive value (PPV) instead of accuracy, which is reflected in their results. Note that there was a large standard deviation for the Hepatitis dataset using the Merlin and Morgan attacks of up to 0.07 making the results difficult to compare. For the Morgan attack, a value close to 0.500 results from the attack identifying a small number of points with $FPR = 0$ and achieving the attack's objective of maximizing PPV. Higher values indicate when this was not possible, and the intended behaviour was then improved by calibration due to the creation of a high precision region e.g. see CIFAR100(aug) and Table 11.

| Dataset | Merlin | | | Morgan | | |
|---|---|---|---|---|---|---|
| | Orig. | Cal. | Cal.-F | Orig. | Cal. | Cal.-F |
| Credit | **0.92** (2.80± 2.04) | 0.71 (1.0±1.26) | 0.84 (2.0±2.60) | 0.64 (0.4±0.8) | 0.80 (0.8±0.75) | **0.92** (1.4±0.8) |
| Hep. | 0.76 (1.0±1.22) | **1.0** (1.0±0.00) | 0.81 (1.0 ± 0.89) | **1.0** (1.5±0.87) | **1.0** (2.75±2.05) | **1.0** (1.0±0.0) |
| Adult | 0.72 (0.2 ±0.40) | 0.76 (1.0 ± 1.55) | 0.77 (0.4 ± 0.49) | 0.78 (0.2±0.4) | **0.80** (0.6±0.49) | **0.80** (1.2± 1.16) |
| MNIST | 0.51 (0.0 ±0.0) | 0.64 (0.0 ± 0.0) | 0.60 (0.0 ±0.0) | 0.52 (0.0±0.0) | 0.64 (0.0±0.0) | **0.66** (0.0±0.0) |
| CIFAR10 | 0.83 (1.2 ±1.60) | 0.79 (0.6 ± 0.55) | 0.78 (0.4 ± 0.55) | 0.83 (1.2 ±1.60) | **1.0** (1.4±0.8) | **1.0** (2.8±2.4) |
| CIFAR100 | **1.0** (2.4 ±1.14) | **1.0** (1.0 ± 0.0) | 0.94 (0.6 ± 0.55) | **1.0** (2.4±1.14) | **1.0** (4.0±1.67) | **1.0** (3.0±1.26) |
| CIFAR10 (aug.) | 0.50 (0.0 ± 0.0) | 0.85 (0.0±0.0) | 0.84 (0.8 ±1.17) | 0.58 (0.0 ±0.0) | **1.0** (1.0±0.0) | **1.0** (1.2±0.4) |
| CIFAR100 (aug.) | 0.50 (0.0 ± 0.0) | **1.0** (2.0±1.55) | **1.0** (1.4±0.49) | 0.72 (0.0 ± 0.0) | 0.98 (1.2 ±0.98) | **1.0** (3.4±1.36) |

Table 11: Positive Predictive Value (PPV) results and number of points identified with $FPR = 0$. In many cases calibration improves PPV. The thresholding technique for the attacks maximized $PPV$, as used by Jayaraman et al. (2021)

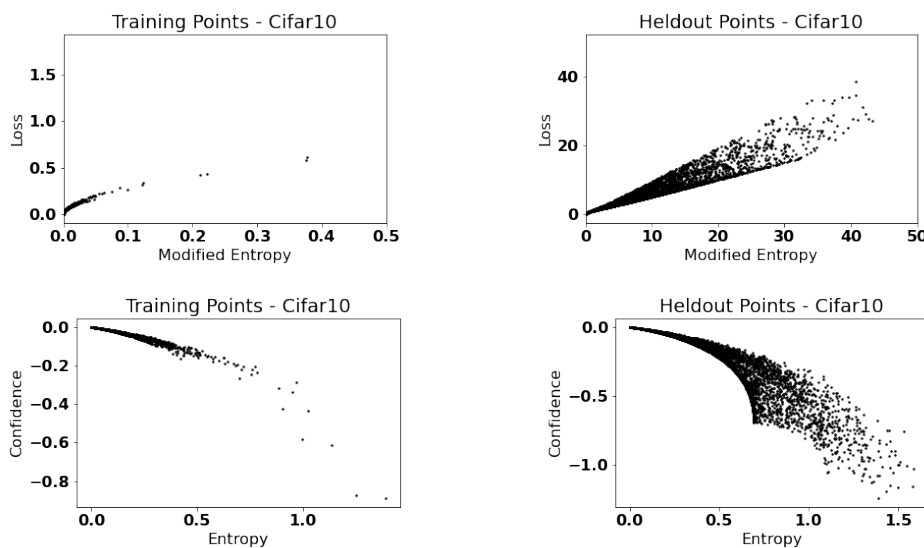

Figure 9: Comparison of entropy and modified entropy with confidence and loss scores respectively for CIFAR10 without data augmentation. Song & Mittal (2021) note that entropy is similar to confidence. Modified entropy is similar to cross entropy loss.

