# OpenReview forum: "On the Importance of Difficulty Calibration in Membership Inference Attacks"
_ICLR.cc/2022/Conference — ICLR 2022 Poster_

### Official Review · Reviewer_32Pd · 2021-11-01

**Correctness:** 3
**Technical Novelty And Significance:** 3
**Empirical Novelty And Significance:** 2
**Recommendation:** 5
**Confidence:** 2

**Main Review:**

Strength: Controlling false positives of membership attacks is important for practical membership inference scenarios. Research efforts along  this direction should be encouraged.

Weakness:
The theoretical rationality of the proposed calibration method is not provided in the study. Though it is mentioned that the proposed method is associated with posterior inference, more formal study establishing the link is still needed. The discussion at the end of Section.3 is not convincing.


**Summary Of The Paper:**

This work proposes a simple output calibration method to suppress false positive errors of membership attacks. The core idea is to define a calibrated decision score as the difference between the original membership score and the expected membership score trained by a randomized training algorithm using a shadow dataset (a dataset following the same distribution as the true training data set). The calibrated membership decision score can provide a lower false positive rate than the original membership score.



**Summary Of The Review:**

The theoretical reasoning the performance of the proposed calibration method is not provided yet necessary.

---

> ### Author Response · Authors · 2021-11-12
> **Response to Reviewer 32Pd**
>
> We thank the reviewer for their comments on our work. We emphasise that our work offers a more practical version of the attacks discussed in Sablayrolles et al. For this reason, we believe that Sablayrolles et al. already provides the theoretical justification for our work.  Our contribution is to extend  this into a practical attack and show via extensive experiments that it can be applied to a wide variety of metric-based attacks to improve their precision-recall tradeoffs. We will make this more explicit in the paper.

---

### Official Review · Reviewer_2TZc · 2021-11-02

**Correctness:** 4
**Technical Novelty And Significance:** 2
**Empirical Novelty And Significance:** 3
**Recommendation:** 8
**Confidence:** 5

**Main Review:**

This paper proposes a natural and simple idea to improve membership inference attacks: train a few reference models on the same distribution, and then use these as a "prior" for the distribution of losses of non-members.
This idea is not entirely new: it was proposed and analyzed both in Sablayrolles et al. and Long et al. which the paper references. It would be helpful to clarify this paper's contribution with respect to these two papers. In particular, it seems to me that the attack proposed in this paper is strictly *weaker* than the one of Sablayrolles et al.
In "Connection to posterior inference", this paper claims that the proposed approach is equivalent to that of Sablayrolles et al. But this isn't quite true: Sablayrolles et al. train reference models on subsamples of the data so that they can approximate both the average loss of an example when it is a non-member *and when it is a member*. Their attack then essentially determines whether an example's observed loss is closer to the average non-member loss or the average member loss of that sample.
The approach in this paper is similar, but only considers reference models that do not contain a victim point. So this attack is necessarily weaker.

While the attack idea thus doesn't seem particularly novel, the analysis and experiments are however very thorough and nicely complement the more theoretical paper of Sablayrolles et al. The ablation studies are overall very solid and give a good idea of what factors influence the attack's strength. A few minor comments:
- Some of the model accuracies in Table 1 are fairly low (e.g., CIFAR-10/100). Do you get similar results if you use a better model or training setup?
- The finding that data augmentation leads to less privacy leakage is contrary to what was found by Choquette-Choo et al. Did you also use augmentations for the attack as in that paper?
- Figure 4 (right) and Figure 6 are hard to read as they display a 3-dimensional relationship. Instead of a full PR curve, it might be better to extract a single scalar for each experiment (e.g., the AUC, or the precision at a recall of 50% or something like that) and then plot that scalar for varying ratios in Figure 4, and for varying epsilon in Figure 6.
- The AUC curves in Figure 5 do not seem to reach a plateau. Can you further improve it by training more reference models?

**Summary Of The Paper:**

This paper proposes to improve membership inference attacks by calibrating the attack threshold on each example's hardness.

**Summary Of The Review:**

A nice paper with an idea that is not entirely novel but with a rigorous empirical analysis.

---

> ### Author Response · Authors · 2021-11-12
> **Response to Reviewer 2TZc**
>
> We thank the reviewer for their thorough and constructive comments. We will respond to your questions and clarify some points below.
>
> *Relation to Sablayrolles et al.:*
>
> Indeed our attack is not equivalent to that of Sablayrolles et al. Instead, we believe their attack motivates ours and that our attack offers a practical way to operationalize its key ideas. As you have pointed out, full equivalence would require retraining for each sample.
>
> *Low model accuracies:*
>
> Our work follows the precedent of using common architectures and accuracies for the CIFAR datasets from  the membership inference literature, for fair comparison with these prior works (Shokri et al., IEEE S&P 2017, Leino and Fredrikson, USENIX 2020). It is indeed the case that  the achieved accuracies are lower than the best performance expected on these tasks using more complex or larger models. However, the Imagenet results in our experiments instead use Resnet18, approaching typical accuracy with a more complex architecture.
>
> *Data augmentation:*
>
> Choquette-Choo et al. perform data augmentation via translations and rotations (by up to 15 degrees). In our work, we used the more standard set of augmentations including random horizontal flipping and random cropping. This difference is the likely explanation for  the different conclusions.
>
> *Plots:*
>
> We will add the suggested AUC plots to improve the readability of these results.
>
> *Number of Calibration Models:*
>
> We agree that it would be interesting to understand the limiting point of adding more calibration models -- we stopped at 10 due to computational resource and time constraints, after seeing diminishing returns.  We will increase the number of models beyond 10 in our next batch of experiments in the coming weeks, and add new observations to the Appendix.

---

### Official Review · Reviewer_QoUg · 2021-11-02

**Correctness:** 3
**Technical Novelty And Significance:** 2
**Empirical Novelty And Significance:** 2
**Recommendation:** 5
**Confidence:** 4

**Main Review:**

[strengths]
1. The idea of avoiding high false positive rates is indeed important for membership inference research.
2. Experimental results on multiple datasets indicate significant improvement than prior approaches.

[weaknesses]
1. The idea of difficulty calibration is not new. As the authors already pointed out, Long et al. (2018) and Carlini et al. (2020) already used similar technique to compare the difference (or ratio) between target models and reference models. What are new things in this paper?
2. The paper fails to compare with existing attacks which are also designed to decrease false positive rates, specifically, the following paper. --- Jayarama et al., “Revisiting Membership Inference under Realistic Assumptions”, PETS 2021.
3. Besides evaluated metrics (loss, gradient norm, confidence), recent works also leverage entropy for membership inference. It would be great if the authors can also include entropy metric as well.  --- Song & Mittal, “Systematic Evaluation of Privacy Risks of Machine Learning Models”, USENIX 2021.


**Summary Of The Paper:**

The paper aims to solve the issue of high false positive rates in modern membership inference attack methods by computing the difference in model prediction metrics (e.g., loss, confidence) between the target model and a reference model. The intuition is that “easy-to-predict non-members” will have a small difference between the target and the reference model, while “hard-to-predict members” will have a large difference. Experimental results on multiple datasets also validate the effectiveness of proposed method.

**Summary Of The Review:**

I like this paper’s focus on reducing high false positive rates. However, unless the authors clearly explain the difference with prior works (Long et al. (2018), Carlini et al. (2020)) and show the advantage over other attacks (Jayarama et al. (2021)), I suggest not accepting this paper.

---

> ### Author Response · Authors · 2021-11-12
> **Response to Reviewer QoUg**
>
> We thank the reviewer for their insightful comments. We will respond to the individual points and make some clarifications below.
>
> *Distinction from previous work:*
>
> Our goal in this work was to provide strong experimental evidence of the efficacy of difficulty calibration that we feel is overlooked by the privacy attack community. Importantly, it does not compete with alternative approaches, but enhances many of them to improve their reliability in practice.  While the idea of comparing to other models is a part of more complex attacks in previous work such as Long et al., we show that the same intuition can be applied more generally to a wide variety of privacy attacks including black-box, white-box and label-only attacks via a very simple modification. Furthermore, our work can be viewed as providing strong empirical evidence for a more practical version of the attack theoretically studied in Sablayrolles et al. We will clarify these points in the paper.
>
> *Compare to Jayarama et al.:*
>
> We thank you for pointing out this interesting recent paper. We reiterate that our goal in this work was to propose a simple modification to existing and future metric-based privacy attacks that improves their precision recall trade-off. For this reason, we do not view our work as in direct competition with that of Jayarama et al. Instead, their attack is an ideal place to apply our proposed method. We plan to evaluate calibrated versions of their newly proposed attack for the camera ready draft if accepted. As Jayarama et al. discuss, their methods are similar in spirit to the label-only attack of Choquette-Choo et al. For this reason, we would expect performance of our method to be similar to that of the confidence based and label-only attacks already included in our results.
>
> *Entropy metric from Song & Mittal:*
>
> Thank you for sharing this reference, which we will discuss in the revised paper.  We will also perform experiments over the next month, and add results to the Appendix, along with the results for calibrated and uncalibrated versions of Jayarama.

---

### Official Review · Reviewer_QzDd · 2021-11-03

**Correctness:** 3
**Technical Novelty And Significance:** 2
**Empirical Novelty And Significance:** 3
**Recommendation:** 5
**Confidence:** 4

**Main Review:**

** Strength **
- The paper is overall well written and easy to understand. The contributions are clearly stated and explained.
- The main contribution is the application of difficulty calibration (inspired by Long et al., 2018) to MIA. Wide body of experimental  results using different datasets (only images though) which mostly support the proposed approach by showing favorable results.
- The analysis with different member-non member ratios is definitely interesting and well done.

** Weaknesses**
* It would be highly beneficial to better and more formally support the intuition about easy-to-predict  non-member samples page 3 section 3. Indeed, the intuition lacks a deep analysis and is only supported by most of the experimental (empirical) results. Fors instance, several fundamental questions remain unsolved such us:
- Are there further connection to the generalization capabilities of the network ? Several results have been mentioned in the literature but this aspect has not been discussed in this work.
- It is not clear of the condition to ensure the validity of expression (6) are satisfied. This assumption should be justified and further investigated.
- What about possible context with the accuracy of the network? Calibration and accuracy are two different concepts but they may also be related to the potential weakness to MIA.


**Summary Of The Paper:**

This paper focuses on the analysis of membership inference attacks (MIAs), in  particular addressing the problem of high false positive rate (FPR) that still affects the  state-of-the-art solutions. The authors suggest an approach to MIA that is based on difficulty calibration, i.e., a method where the predicted membership score is adjusted  to the difficulty of obtaining the correct classification for the target sample, in order to reduce the FPR. As illustrated in Figure 1, previous work on the score-based membership inference attack is highly unreliable for separating easy-to-predict non-members from hard-to-predict members, as both can achieve high membership scores. This technique appears to be particularly beneficial in helping to distinguish between  member and non-member samples that are overrepresented in the data distribution. The authors support the proposed method with a large body of mostly favorable experimental results.

**Summary Of The Review:**

In order to better understand the improvement coming with the proposed method, it would be useful to have a deeper understanding of the reason for which it is indeed beneficial. More details on the differences between the models f and g are to be provided: different random
initializations are probably involved as well as different batch shuffling. I guess that mainly the g models are going to converge faster on the training data, presenting more variance in the performance for non member data, which in turn makes it easier to find an effective threshold for the MIA problem. However, this does not seem effective on some datasets. Could the authors investigate those issues and shed more light on the relation between the type of data and the success/insuccess of their method?

---

> ### Author Response · Authors · 2021-11-12
> **Response to Reviewer QzDd**
>
> We thank the reviewer for their thorough comments. We will provide clarifications below.
>
> *Different datasets:*
>
> Please note that we include both image and non-image datasets in our experiments. For example, the Adult and Credit datasets are not image datasets.
>
> *Formal support for the intuition:*
>
> We view our proposed methods as a more practical version of attacks theoretically studied in Sablayrolles et al. Therefore, we believe that their results provide a theoretical foundation for our work. We will clarify this in the paper.
>
> *Are there further connections to the generalization capabilities of the network?*
>
> Generalization measures loss(model, x_1) - loss(model, x_2) and we look at loss(model_1, x) - loss(model_2, x). We believe that the two quantities should behave fairly similarly. However it is possible to imagine scenarios in which the two quantities do not have the same properties. For instance, if the training set is made up only of “difficult” examples but that these examples still enable the model to learn well, then loss(model, x_1) - loss(model, x_2) will be low (the model has “memorized” x_1 and generalizes well to x_2) but loss(model_2, x) - loss(model_1, x) will be high because model_2, not having seen “x”, will give it a high loss.
>
> *Justification of Equation 6:*
>
> There are a variety of ways to justify Equation 6. For example, in the case of models trained using SGD with added gaussian noise, Welling & Teh (ICML 2011) show the iterates converge to Bayesian posterior samples from the distribution in Equation 6 of our paper.  Sablayrolles et al. also provide justification in Section 3.1. We will add a discussion of this to ensure that it is clear.
>
> *Relation to accuracy:*
>
> The comment regarding calibration and accuracy is unclear to us, can you elaborate?
>
> *Training details:*
>
> We will clarify that each calibration model was randomly initialized and that the batches were randomly shuffled.
>
> *Relation to types of data*
>
> Our results indicate that the difficulty and complexity of the task is related to the performance of difficulty calibration. In particular, difficulty calibration identifies memorized outlier samples with high precision, hence benefitting from a high variance in the underlying data distribution. MNIST is an easier task that is difficult to attack. The MNIST data does not have particularly high variance and the generalization ability of a model trained using a subset of the data is high. In contrast, CIFAR100 is a harder task, with significantly more variation in the underlying data and a much lower generalization ability. Difficulty calibration is more effective in this setting.

---

> > ### Comment · Reviewer_QzDd · 2021-11-28
> > **Acknowledgement of authors' response and efforts in reviewing the paper.**
> >
> > Some of my questions have been clarified. Others like the ones of relation to accuracy and  further connections to the generalization capabilities of the network still remain open. By relation to accuracy, I was referring to the intrinsic different that arises between learning to imitate nature compared to learning nature itself. The former is related to the accuracy of the model while the later is related to learning the posterior distribution of concepts given the features. How is the second problem related to the calibration of the model and the MIA vulnerability  ?

---

> > > ### Author Response · Authors · 2021-12-02
> > > **Response to Reviewer's Comment**
> > >
> > > We agree that adding more discussion of the relationship between calibration, membership inference attacks and training/generalization performance will further strengthen the paper.
> > >
> > > *Generalization*:
> > >
> > > To extend the point we made in our response, we think that membership inference is fundamentally a problem of generalization in the following sense: the model trained on a sample (x, y) will slightly overfit on it. Thus, if this model sees x, it will predict the label y with more confidence than a model not trained on (x, y). This gap is what enables us to perform membership inference with high precision. Note that people usually consider generalization as the average of the gap between the loss on “in” and the loss on “out” samples. This average can be low which hides the fact that for some samples the gap is high (these samples are outliers / overfitted) and for the major part of the distribution the gap is low (these samples are inliers / not overfitted). This is why we advocate for precision/recall rather than accuracy: knowing with 100% certainty that 1% of the samples were in the training set is better than knowing with 60% certainty that 5% of the samples were in the training set.
> > >
> > > We will add this discussion to the paper.
> > >
> > > *Accuracy:*
> > >
> > > We will also discuss how the training accuracy of the model can affect both the performance of membership inference attacks and of calibration. Given a fixed generalization accuracy, models with higher training accuracies are more prone to overfit and therefore more vulnerable to membership inference attacks, as discussed above. Calibration will affect them more strongly as the comparison between the overfit model being attacked and the ‘average’ model will be more pronounced. We can see evidence of this in our empirical results. On the other hand, if training accuracy is so low that no meaningful fitting of the model has taken place, then both membership inference attacks and calibration will be unsuccessful.

---

### Decision · Program_Chairs · 2022-01-20

**Decision:**

Accept (Poster)

**Comment:**

This paper proposes a technique to improve membership inference attacks by
carefully applying "difficulty calibration" to improve the attack success
rate. The reviewers are split on this paper. They all generally agree on
the facts: the paper introduces a (somewhat) new technique and performs a
solid evaluation, but the novelty on top of prior work isn't all that high.

On the whole I believe this paper should be accepted. This paper has identified
a very clear problem with existing attacks (poor performance at low false
positive rate) and has carefully developed a way to improve on this metric. A
thorough evaluation has convinced the reviewers that this paper does what it
set out to do.

It comes down to a question of novelty then. And here the question is this:
does someone who reads this paper learn something new that wasn't obvious
before? Part of this can be novelty in the method---and I agree with the
reviewers that this paper lacks novely in the method. However the paper does
not lack novelty in the ideas on the whole. While Long et al., Sablayrolles et
al., and Carlini et al. do all use some kind of low false positive rate
evaluation and calibrate for low loss, none of these papers actually go out
of their way to evaluate this fact explicitly. And so even if this paper
had no technical contribution at all, the simple measurement study in and
of itself would be a useful insight.

Machine learning research at present focuses fairly heavily on novelty
of the techniques. While this is good, it's also important to go back and
actually evaluate what we have. That's what this paper does, and it does
it well enough to be worth accepting.

The paper would definitely be improved by following some of the advice of
the reviewers and including comparisons to prior work (e.g., especially
clarifying the relationship to Sablayrolles et al. and if it is true that
this attack is a simplification of this prior one and is thus less effective)
and I hope the authors will take the opportunity to do this.